# Universal control of a bosonic mode via drive-activated native cubic interactions

**Axel M. Eriksson** [1] ✉, **Théo Sépulcre** [1], **Mikael Kervinen**[1], **Timo Hillmann** [1], **Marina Kudra**[1], **Simon Dupouy** [1], **Yong Lu**[1,2], **Maryam Khanahmadi** [1], **Jiaying Yang** [1], **Claudia Castillo-Moreno** [1], **Per Delsing** [1] & **Simone Gasparinetti** [1] ✉

Linear bosonic modes offer a hardware-efficient alternative for quantum information processing but require access to some nonlinearity for universal control. The lack of nonlinearity in photonics has led to encoded measurement-based quantum computing, which relies on linear operations but requires access to resourceful ('nonlinear') quantum states, such as cubic phase states. In contrast, superconducting microwave circuits offer engineerable nonlinearities but suffer from static Kerr nonlinearity. Here, we demonstrate universal control of a bosonic mode composed of a superconducting nonlinear asymmetric inductive element (SNAIL) resonator, enabled by native nonlinearities in the SNAIL element. We suppress static nonlinearities by operating the SNAIL in the vicinity of its Kerr-free point and dynamically activate nonlinearities up to third order by fast flux pulses. We experimentally realize a universal set of generalized squeezing operations, as well as the cubic phase gate, and exploit them to deterministically prepare a cubic phase state in 60 ns. Our results initiate the experimental field of polynomial quantum computing, in the continuous-variables notion originally introduced by Lloyd and Braunstein.

Quantum information processing relies on the capability to operate quantum superpositions between several quantum states in a large system. A mainstream approach to quantum computing is based on ensembles of coupled two-level systems[1,2]. However, an alternative approach is to utilize bosonic modes[3–5], also called continuous-variables (CV) modes. Each bosonic mode directly gives access to a large Hilbert space of quantum states, which could be used to redundantly encode and protect quantum information from errors in a hardware-efficient manner[3,6–10], or, in principle, implement CV quantum computing[11]. However, to operate a CV mode, it is crucial to introduce a nonlinearity that allows one to universally manipulate the superpositions of the quantum states[4].

On the photonics platform, the main obstacle to quantum computing is the weakness of the accessible nonlinearities (relative to the intrinsic losses) in optical crystals. A common strategy in photonics is, therefore, to use readily accessible Gaussian operations such as squeezers and beamsplitters in combination with non-Gaussian measurements, such as those performed by single photon detectors[12,13]. One approach to universal control is to utilize resourceful non-Gaussian input states[14–16], such as the cubic phase state, to realize non-Gaussian gates by gate teleportation[3,17].

In contrast to the challenges in photonics, nonlinearities for CV modes in superconducting circuits are readily available, originating from Josephson junctions that act as nonlinear, low-loss inductive elements[18]. These CV modes can be constituted by superconducting cavities weakly (dispersively) coupled to a qubit providing universal controllability[19–22]. For the superconducting modes, the problem compared to the photonic domain is reversed since the linear CV

[1]Department of Microtechnology and Nanoscience, Chalmers University of Technology, 412 96 Gothenburg, Sweden. [2]Physikalisches Institut, University of Stuttgart, 70569 Stuttgart, Germany. ✉e-mail: axel.eriksson@chalmers.se; simoneg@chalmers.se

modes inherit a static, always-on Kerr nonlinearity from the hybridization with the Josephson junction[23], which limits the fidelity of operations. Weakening the dispersive interaction reduces the inherited nonlinearity but also leads to slower interactions which are typically on the order of a microsecond[21,22]. Furthermore, despite considerable advances in superconducting qubits, the variety of noise channels introduced by the ancilla control qubit still limits the system performance[9,24].

However, a strength of superconducting devices is the possibility of tailoring and in situ tuning the nonlinearities with the external magnetic flux through superconducting loops. Especially by arranging Josephson junctions into an asymmetric loop as in a superconducting nonlinear asymmetric inductive element (SNAIL)[25], the Josephson potential becomes asymmetric. Hence, suitable tuning of the external flux enables the cancellation of the static Kerr nonlinearity while maintaining a strong third-order nonlinearity, making the SNAIL useful in quantum amplifiers[26,27] and in reducing the static interaction in qubits[28] and linear couplers[29].

In this paper, we demonstrate universal control of a bosonic mode in a superconducting circuit comprising a planar resonator terminated by a SNAIL[30] without an ancilla control qubit. The nonlinear controllability of the CV mode instead originates from intrinsic nonlinearities within the bosonic mode, which primarily manifests themselves when driven. Hence, different nonlinear interactions are turned on only when certain drive pulses are applied (Fig. 1a). These nonlinearities are realized by embedding a SNAIL into the bosonic mode resonator itself (Fig. 1b, c). By flux and charge driving the SNAIL-terminated resonator, we activate Gaussian and non-Gaussian (nonlinear) interactions to implement fast universal quantum gates within tens of nanoseconds. In contrast to previously demonstrated bosonic gate sets[20,22], the gate set implemented here is a natively polynomial set, including the non-Gaussian cubic gate. We thereby implement a universal gate set in the

notion of CV quantum computing inspired by the original proposal by Lloyd and Braunstein[11]. Furthermore, by leveraging the native cubic gate, we experimentally generate a cubic phase state with enhanced speed and fidelity compared to our previous work[21] in 3D cavities.

## Results

### Bosonic mode with drive-activated nonlinearities

We implement our device on a superconducting planar architecture fabricated with conventional lithography techniques and measured at ~10 mK in a dilution refrigerator (see Methods). We realize the bosonic mode with drive-activated nonlinearities by terminating a $\lambda/4$ resonator with a SNAIL element at its current anti-node (micrograph in Fig. 1d). A transmon qubit[31] is dispersively coupled to the bosonic mode as well as to a readout resonator to perform Wigner tomography of the bosonic state[20]. Crucially, the qubit does not participate in the control of the bosonic mode. Importantly, the SNAIL element alters the boundary condition of the otherwise linear mode and thereby introduces the nonlinearities that will be utilized for the control. The flux tunability of the SNAIL potential results in a nonlinear frequency tuning of the mode with respect to the external magnetic field through the SNAIL flux loop, as measured via the cavity Ramsey protocol[32] (Fig. 1e).

The nonlinear Hamiltonian of the SNAIL-terminated resonator[30], driven by charge and flux, can be expanded around the minimum of the potential well and takes the form

$$\hat{H}/\hbar = \omega_0 \hat{a}^\dagger \hat{a} + \xi(t)\left(\hat{a}^\dagger + \hat{a}\right)$$
$$+ \sum_{n=1}^{\infty} \left(g_n^{dc}(\phi_e^{dc}) + g_n^{ac}(\phi_e^{dc})\phi_e^{ac}f(t)\right)\left(\hat{a}^\dagger + \hat{a}\right)^n \tag{1}$$

where $g_1^{dc} = g_2^{dc} = 0$ and with frequency $\omega_0$, bosonic creation operator $\hat{a}^\dagger$, charge drive amplitude $\xi(t)$, and linear/nonlinear coefficients $g_i^j$ for

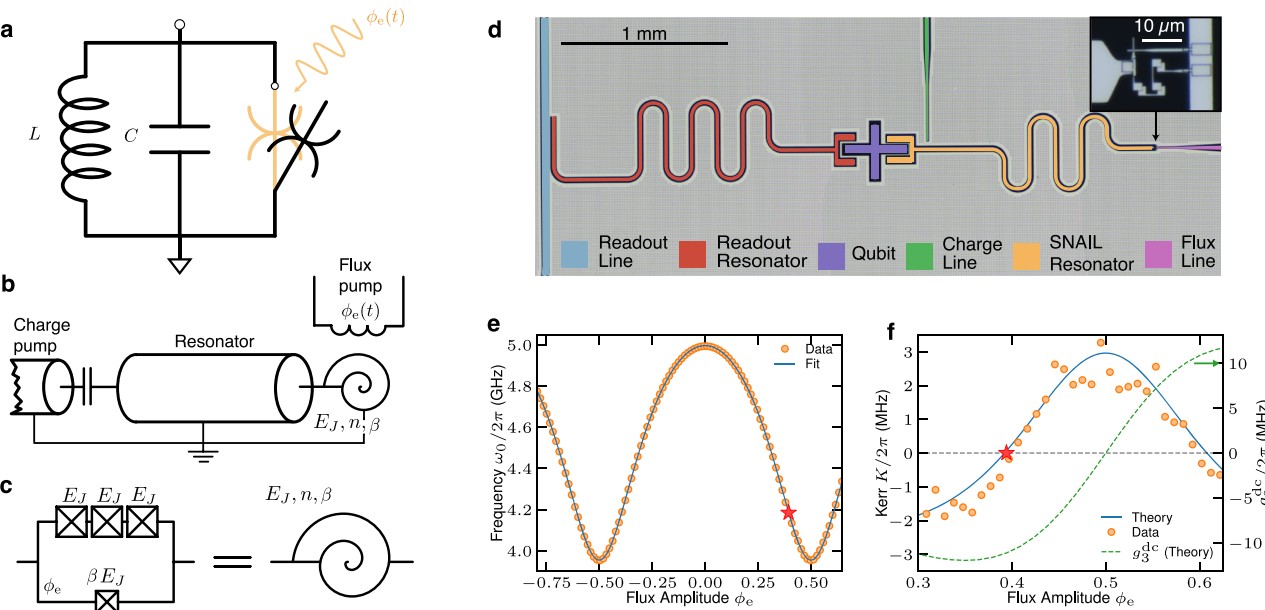

**Fig. 1 | Bosonic mode with drive-activated nonlinearities. a** Idealized circuit schematic. In the undriven state (open black symbol), the resonator behaves as a harmonic oscillator, characterized by its inductance $L$ and capacitance $C$. Manipulation of the quantum states is achieved by driving the resonator with flux $\phi_e(t)$, which activates the (otherwise off-resonant) intrinsic nonlinearities. Different pulse compositions engage (closed orange symbol) a variety of interactions denoted by the switchable, purely nonlinear junction element. **b** Schematic of our circuit realization consisting of a SNAIL-terminated $\lambda/4$ coplanar waveguide resonator, following and adopting figure from ref. 30. **c** Schematic of the SNAIL element with $n = 3$ Josephson junctions with Josephson energy $E_J$ and asymmetry factor $\beta$. **d** False-

colored micrograph of the full device layout. The SNAIL-resonator is equipped with a charge and a flux drive line and dispersively coupled to a transmon qubit with a readout resonator. (inset) Close-up of the SNAIL element with three large Josephson junctions on one arm and a single junction on the other arm. **e** SNAIL-resonator frequency tuning vs static flux. The solid line is the fitted model taking into account the relevant microscopic parameters (see Methods). **f** Tuning of resonator nonlinearities vs static flux; fourth order (left axis) and third order (right axis). Effective Kerr $K^{(1)}$ and $g_3^{ac}$ nonlinearities are predicted by the model fitted to the frequency tuning. The bosonic mode with drive-activated nonlinearities is realized and operated at the Kerr-free point marked with a star.

$i \in \mathbb{N}$ and $j$ spanning both static dc terms as well as flux ac components, which depend on the static flux $\phi_e^{dc} = \Phi_e^{dc}/\Phi_0$, $\Phi_0 = h/2e$. The flux drive field is constituted by the flux amplitude $\phi_e^{ac}$ and normalized pulse shape $f(t)$, where $\max(f(t)) = 1$. By fitting the microscopic parameters of the static components (see Methods) to the measured nonlinear tuning of the frequency as a function of static magnetic flux (Fig. 1e), we predict the strengths of the nonlinearities $g_i^j$ (Fig. 1f). The predicted effective Kerr is validated by measuring the same via the out-and-in protocol (see Methods). All fitted and characterized parameters are summarized in a table in the Supplementary Table 1.

Importantly, the magnitudes of the effective static Kerr nonlinearities are strongly suppressed at flux $\phi_e^{dc} = 0.3930$, creating a Kerr-free region for phase space coordinates $|\alpha| \lesssim 1.5$ (see Methods). This cancellation is a key feature of the SNAIL, which is possible due to its asymmetric design allowing for the cancellation of one arbitrary order in the Taylor expansion of the potential[25]. Therefore, we choose this (close to) Kerr-free point as the operating point of our device, characterized by a resonant frequency $\omega_0/2\pi = 4.158$ GHz, energy relaxation time $T_1 = 28$ μs, and cubic nonlinearity $g_3^{ac}/2\pi \sim -10$ MHz. Hence, when no drives are applied, the resonator behaves approximately as a harmonic oscillator. We achieve quantum control of the cavity state by sending microwave pulses to on-chip flux and charge lines. By judiciously choosing the frequencies and composition of the driving pulses, we activate a range of linear and nonlinear interactions, as demonstrated below.

## Universal interactions via flux driving

The notion of universality introduced by Lloyd and Braunstein for continuous-variable systems[11] requires a set of Gaussian gates and at least one non-Gaussian gate, which can be chosen arbitrarily among the polynomials of degree three or higher in the quadratures of the bosonic modes. Hence, we experimentally obtain a universal gate set

by flux driving the SNAIL in such a way that each gate consists of a single monochromatic pulse at a multiple of the resonator frequency. We demonstrate the gate set by flux pumping the SNAIL at one, two, and three times the resonance frequency $\omega_0$, which resonantly engage the displacement, squeezing (Fig. 2a–c) and trisqueezing[33] (Fig. 2d–f) interactions, respectively. Together with the trivial rotation, these interactions constitute the universal generalized-squeezing gate set

$$\hat{R}(\theta) = e^{-i\theta \hat{a}^\dagger \hat{a}} \tag{2}$$

$$\hat{D}(\alpha) = e^{\alpha \hat{a}^\dagger - \alpha^* \hat{a}} \tag{3}$$

$$\hat{S}(\zeta) = e^{(\zeta \hat{a}^{\dagger 2} - \zeta^* \hat{a}^2)/2} \tag{4}$$

$$\hat{T}_s(\tau) = e^{\tau \hat{a}^{\dagger 3} - \tau^* \hat{a}^3}. \tag{5}$$

with rotation $\hat{R}(\theta)$ by an angle $\theta$, displacement $\hat{D}(\alpha)$, squeezing $S(\zeta)$ and trisqueezing $T_s(\tau)$, where the complex parameters $\alpha$, $\zeta$, and $\tau$ describe the magnitude and phase of the operations. The magnitude of $\alpha$, $\zeta$, and $\tau$ are to first order proportional to the Hamiltonian parameters $g_1^{ac}$, $g_2^{ac}$ and $g_3^{ac}$ in Eq. (1), respectively (see Methods). The ability to engage these processes from Eq. (1) can be intuitively understood by the drive field $f(t) \propto \cos(n\omega_0 t)$, resonantly capturing interactions $\hat{a}^n$ and $\hat{a}^{\dagger n}$ at different orders $n$. Crucially, the trisqueezing interaction is non-Gaussian, promoting the otherwise Gaussian gate set to a universal gate set. By engaging the trisqueezing gate, we produce the non-gaussian Wigner-negative[15] trisqueezed state (Fig. 2e). The quantum states are measured by direct Wigner tomography via the dispersively coupled spectator qubit. The squeezing and trisqueezing levels can be enhanced until ~9 and -0.8 dB, respectively, above which the simulated

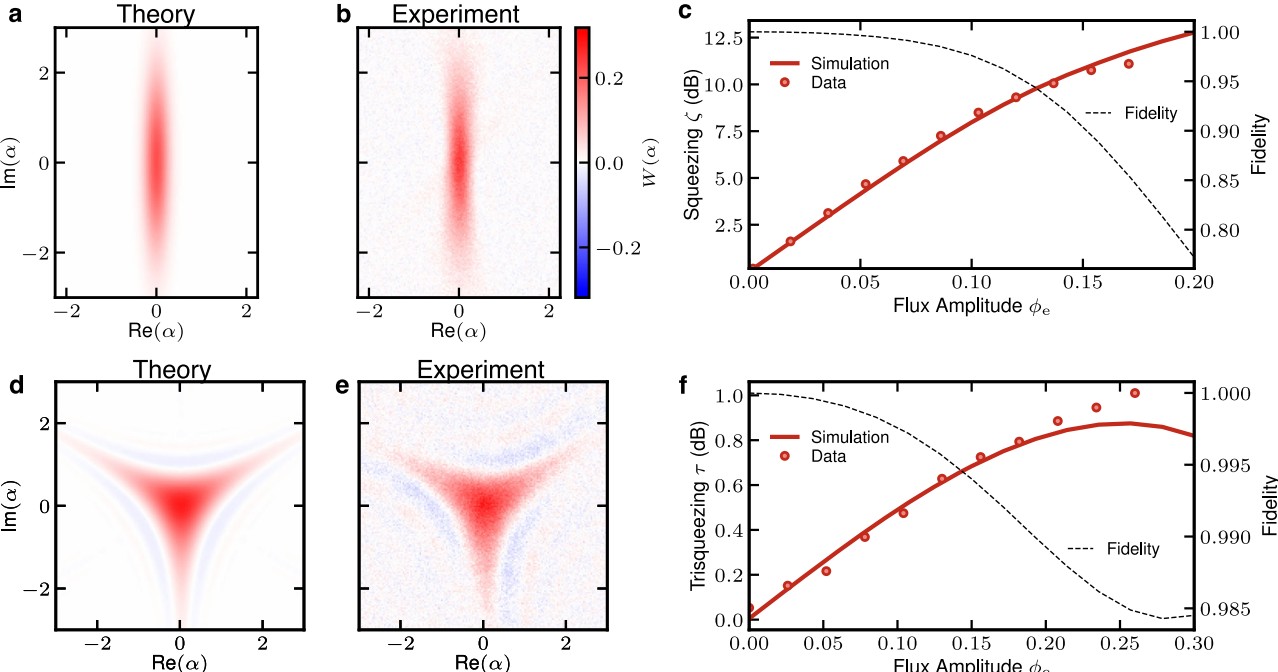

**Fig. 2 | Universal control of the SNAIL-resonator achieved by monochromatic flux pulses.** Results are shown for squeezing (top panels) and trisqueezing (bottom panels). A 20 ns $2\omega_0$ pulse parametrically activates the squeezing interaction and squeezes the cavity vacuum state to $\zeta = -0.99$ ($-8.63$ dB). The resulting Wigner state is fitted with a pure squeezed state (**a**) via the Wigner overlap to the measured state (**b**). **c** Increasing squeezing level with flux pulse amplitude. Pulse duration also increases the squeezing level but is here fixed to 40 ns. Data points are obtained by

fitting the measured Wigner functions to the corresponding pure states via Wigner overlap. Simulated state squeezing level (solid) and fidelity (dashed). Similarly, a 60 ns $3\omega_0$ flux pulse activates the trisqueezing interaction, which trisqueezes the cavity vacuum state into a Wigner negative trisqueezed state fitted (**d**) and measured (**e**) with $\tau = -0.13$. **f** Increasing trisqueezing level analogous to the squeezing case.

fidelities start to drop because of higher-order nonlinearities. Furthermore, these flux-activated gates are executed within tens of nanoseconds. This is to be contrasted with bosonic gates assisted by a dispersively coupled qubit, such as selective number-dependent arbitrary phase (SNAP) gates[20,21] or optimal control pulses[34], which are 10–100 times slower due to the weakness of the dispersive interaction.

We emphasize that the gates demonstrated in Fig. 2 are performed by driving flux tones, whereas experimental demonstrations for superconducting 3D modes often are limited to charge driving because of the difficulty to bring ac-flux fields into the superconducting cavities[35]. This enables us to use interactions via both the linear charge and parametric flux paradigm to tailor a variety of gates (see Methods for illustrative diagrams). As an example, a trisqueezing gate could, in principle, be performed solely with charge driving at $\omega_d = 3\omega_0$ in the presence of a $g_4^{dc}$ interaction term. But this interaction also triggers a static Kerr effect. Tuning the system to reach the Kerr-free-flux point is not beneficial in this context, as it also cancels the amplitude of trisqueezing gate. By contrast, the use of a parametric drive $g_3^{ac}$ does provide the trisqueezing gate at the Kerr free point without any residual interaction when idling. Similarly, each $g_n^{ac}$ in Eq. (1) allows one to activate an $n$-photon process by flux driving resonantly at $n\omega_0$, with greatly suppressed unwanted static effects.

### Demonstration of the cubic phase gate

Following the proposal by Hillmann et al.[30], we utilize simultaneous flux and charge drives to natively implement the cubic phase gate (Fig. 3),

$$\hat{C}(\gamma) = e^{i\gamma\left(\frac{\hat{a}^\dagger + \hat{a}}{\sqrt{2}}\right)^3} \tag{6}$$

with cubicity parameter $\gamma$. The cubic interaction contains the cross terms proportional to $\hat{a}^\dagger\hat{a}^2$ and $\hat{a}^{\dagger 2}\hat{a}$, which distinguishes it from the trisqueezing gate. To activate the full cubic interaction (Fig. 3a), we, therefore, simultaneously apply a $3\omega_0$-flux pulse which captures the trisqueezing interaction, and a $1\omega_0$-flux pulse capturing the cross terms. However, since the $1\omega_0$-flux pulse also engages direct displacement, we concurrently apply a $1\omega_0$-charge pulse to cancel the linear displacement. Since the displacement interaction is much stronger than the cross interaction ($g_1^{ac}/g_3^{ac} \approx 2100$), the cubic phase gate requires precise experimental calibrations to balance the timing, strength, and angles of the drive pulses (see Methods).

Finally, we combine the squeezing gate with the cubic gate to deterministically generate a cubic phase state from the vacuum state. We generate the resourceful[36] cubic phase state $|\zeta,\gamma\rangle = \hat{C}(\gamma)\hat{S}(\zeta)|0\rangle$

from the vacuum state $|0\rangle$ by first applying a 20 ns squeezing gate followed by a 40 ns cubic gate. This sequence results in a cubic phase state (Fig. 3b, c) characterized by $\zeta = -0.61$ (corresponding to 5.3 dB of squeezing) and $\gamma = 0.11$. The fidelity to the closest pure cubic phase state is estimated to be 92 % from generative adversarial neural network reconstruction[37] (see Methods). Importantly, the cubicity of the cubic phase state showcased in this study can be continuously increased by simply employing a longer cubic gate. Being able to enhance the cubicity via the cubic gate is in striking contrast to the prior demonstration of a cubic phase state[21], in which selective-number arbitrary phase (SNAP) and displacement gates had to be re-optimized for every new cubicity value.

## Discussion

We construct an error analysis using numerical simulations to examine the factors that restrict the performance of cubic phase state generation (see Methods). The analysis reveals that the fidelity of the generated state is primarily limited by the coherence time of the SNAIL resonator ($T_2^* = 2.8\,\mu s$ dominated by $1/f$ noise, see Methods), followed by residual thermal population in the SNAIL-resonator ($n_{th} = 2.4\%$). By contrast, simulations indicate that the higher-order corrections $g_5^{dc}$ and $g_6^{dc}$ have a very limited influence on the fidelity of the produced cubic phase state, whereas they become a dominant source of error for larger states. In future realizations, the flux sensitivity of the SNAIL at the operational Kerr-free point could potentially be improved by optimizing the SNAIL parameters[29] without compromising the strengths of the desired nonlinearities. Another alternative is to optimize the tradeoffs in an ATS element[38]. An improved fabrication process is expected to extend the cavity lifetime[39]. The thermal population could be further lowered by improving the filtering of the flux line (see Supplementary Note 1). A single SNAIL could be replaced by an array of $m$ SNAILs, for which higher-order nonlinearities at order $n$ are suppressed as $g_n(t) \sim m^{1-n}$. Care must be taken to ensure the homogeneity of both the SNAILs within the array and the flux applied to each SNAIL. Simulations also indicate that there is room to drive the device harder before other unwanted nonlinear effects occur. Faster gates could then be obtained by removing the hardware limitations that prevented us from further increasing the amplitude of the driving tones in the present experiment. Finally, we observe a systematic flux drift in the data batches of the cubic phase state. The drift manifests itself as a tilting cubic phase state, which we ascribe to the system drifting away from the Kerr-free point. Hence, we expect more frequent system calibration to improve the state generation performance.

In conclusion, we have experimentally demonstrated a polynomial gate set for a bosonic mode in the notion proposed by Lloyd

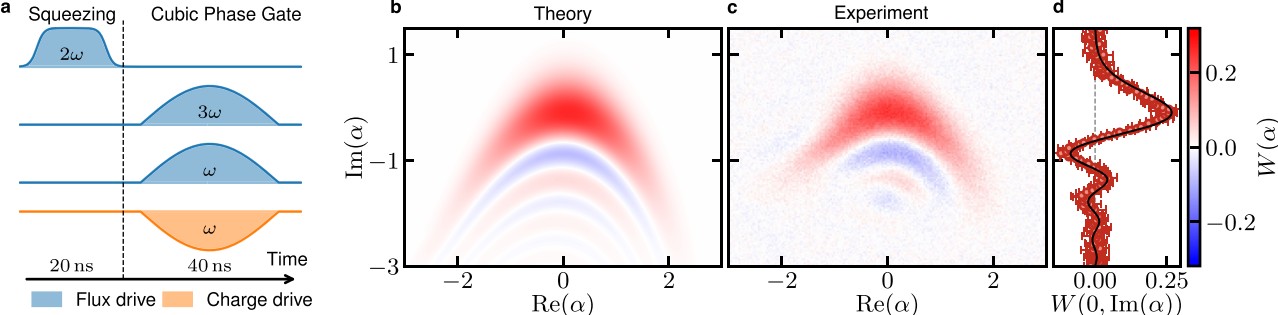

**Fig. 3 | Experimental demonstration of the cubic gate by generation of a cubic phase state in 60 ns. a** Pulse sequence. First, we apply a squeezing gate to the vacuum by a 20 ns $2\omega_0$ flux pulse and then a cubic gate constituted by a $3\omega_0$ flux pulse, a $1\omega_0$ flux pulse and a counteracting $1\omega_0$ charge pulse, at the same time, for 40 ns. **b** Fitted and **c** measured Wigner function of the generated cubic phase state with squeezing $\zeta = -0.61(5.3\,dB)$ and cubicity $\gamma = 0.11$ and a residual displacement $\alpha = -0.090 + i0.025$. **d** Line cut of the Wigner function in (**c**) at $Re(\alpha) = 0$: experimental data (symbols) with 1 std. dev. error bars, and corresponding theory prediction for the state in (**b**) (solid line).

and Braunstein[11]. Hence, our platform initiates the field of continuous-variable-noisy intermediate-scale quantum (CV-NISQ) polynomial quantum computing[40] and simulations[41]. We envision utilizing the SNAIL-resonator as a specialized computational unit interfacing with longer-lived bosonic modes (for example, via beamsplitter interactions[29]), either to rapidly prepare highly nonclassical states or to execute highly nonlinear gates. In addition, our demonstration of a strong, native trisqueezing gate spurs the question of how continuous-variable algorithms could directly benefit from such a resource. The tunable nonlinearities in the SNAIL-resonator further allow for switching its functionality in situ to operate in the Kerr-cat regime[42–44] and even investigate higher order time crystals[45,46]. Another direction would be to combine the intrinsic polynomial interactions with the control offered by an ancilla qubit via optimized selective number-dependent arbitrary phase gates[21] or full optimal control[20]. Finally, if the fast progress on efficient microwave-to-optics transducers[47] continues, one can envision upconverting cubic phase states produced on the microwave platform to optical frequencies. That is, generating the resource states required for fault-tolerant, measurement-based quantum computing on a hybrid superconducting-photonics platform.

## Methods

### Estimating microscopic parameters

The Hamiltonian used to represent the SNAIL resonator in Eq. (1) is written in terms of parameters $\omega_0$, $g_i^{ac}$ and $g_i^{dc}$. It would be difficult to extract a value from each of them independently. Instead, we express them as functions of a few relevant microscopic parameters of the underlying circuit, which are obtained all at once by fitting the $\omega_0 = f(\phi_e)$ experimental data. We now describe these microscopic parameters.

The SNAIL element is described in Fig. 1d. We denote $\varphi$ the superconducting phase at one of its node, the other being grounded. If we assume our probing frequency negligible compared to the plasma frequencies of the SNAIL junction array, we can model it as a nonlinear potential imposed on $\varphi$:

$$U(\varphi) = -E_J\left(\beta\cos(\varphi) + n\cos\left(\frac{\phi_e - \varphi}{n}\right)\right), \quad (7)$$

where $\phi_e$ is the external flux piercing the SNAIL loop (from now on, $\hbar = 2e = 1$). The potential minimum $\varphi_m$ is given by $\beta\sin(\varphi_m) = \sin((\phi_e - \varphi_m)/n)$. Expanding $U$ around this minimum as a Taylor expansion, $U = \sum_{n=2}^{\infty}\varphi^n d^n U/d\varphi^n(\varphi_m)/n!$. All terms are non-vanishing, contrary to what would happen for a SQUID element.

The bare resonator (without the SNAIL) is characterized by its lowest mode frequency $\omega_\infty$ and impedance $Z$. The quadratic term in the Taylor expansion will act like a boundary inductor, shifting the resonance frequency to $\omega_0$. The SNAIL flux is expanded in the resonator creation/annihilation operators as $\hat\varphi = \Phi(\hat a^\dagger + \hat a)$, with proportionality factor $\Phi$, such that the resonator Hamiltonian reads

$$\hat H = \omega_0\hat a^\dagger\hat a + \sum_{n=3}^{\infty}\frac{\Phi^n}{n!}nU\varphi\varphi_m(\hat a^\dagger + \hat a)^n. \quad (8)$$

See Supplementary Note 2 for expressions of $\omega_0$ and $\Phi$ as functions of the microscopic parameters. $\omega_0$ is measured via a Ramsey interference protocol. It is fitted against our microscopic model with 5 free parameters, $\beta$, $\omega_\infty$, $Z/E_J$, and external flux calibration with a linear model, $\phi_e^{dc} = V^{dc}/V_0^{dc} + \phi_{offset}^{dc}$, with voltage $V^{dc}$ applied to the flux source, voltage corresponding to one flux quanta $V_0^{dc}$ and the flux offset $\phi_{offset}^{dc}$. The fitted curve is represented in Fig. 1e, and the fit parameter values are given in Supplementary Table 1. $E_J$ is obtained independently by normal state resistance measurement.

### AC and DC nonlinear terms

The external flux is modulated around its DC value as $\phi_e(t) = \phi_e^{dc} + \phi_e^{ac}f(t)\cos(\omega_d t)$, where $f(t)$ is a slowly varying pulse envelope with $\max f(t) = 1$, which drives the system. Accordingly, we separate the coefficients in Eq. (8) in their DC and AC parts. We expand the Hamiltonian coefficients at first order around $\phi_e^{dc}$ (assuming $\phi_e^{ac} \ll 1$) and obtain Eq. (1), where $g_1^{dc} = g_2^{dc} = 0$ and

$$g_n^{dc} = E_J\frac{\Phi^n}{n!}\frac{\partial^n U}{\partial\varphi^n}\bigg|_{\substack{\phi_e^{dc}\\\varphi_m}}, \quad g_n^{ac} = E_J\frac{\Phi^n}{n!}\frac{\partial}{\partial\phi_e}\frac{\partial^n U}{\partial\varphi^n}\bigg|_{\substack{\phi_e^{dc}\\\varphi_m}}. \quad (9)$$

Expressions for the derivatives can be evaluated explicitly. Note that nonlinear coefficients $g_n^{dc}$ and $g_n^{ac}$ are completely determined by previously fitted parameters $\omega_\infty$, $\alpha$, $Z$, and $E_J$, as done in Fig. 1f.

### Numerical simulations

We systematically compare the performances of our experimental system with numerical simulations of its dynamics. The simulations are done with the QuTiP[48] Python library, using time integration of the Lindblad equation of motion. We include all drives with their pulse shapes as nonlinear terms up to order 6 in Eq. (1), energy relaxation, and dephasing, all according to the numerical values obtained from experimental data summarized in Supplementary Table 1. We assume the initial cavity state and the environment to be in thermal equilibrium at 50 mK, a value obtained by measuring the residual excited population of the SNAIL resonator. These simulations are used in Fig. 2c, f, where they provide a theory to experiment agreement beyond the linear regime at weak drive.

### Kerr-free point

The rotating wave approximation (RWA) is a customary method to study driven nonlinear oscillators[49], which neglects fast rotating terms in the oscillator frame. The method can be systematically extended in a perturbative expansion in $g_n/\omega_0$[50–52]. In the absence of driving, the effective Hamiltonian contains only number-conserving terms:

$$\hat H_{eff} = (\omega_r - \omega_0)\hat a^\dagger\hat a + \sum_{n=1}^{\infty}\frac{K^{(n)}}{n+1}\hat a^{\dagger n}\hat a^n, \quad (10)$$

where $\omega_r$ is the renormalized oscillator frequency, $K^{(1)}$ accounts for Kerr effect, and the $K^{(n>1)}$ for higher order effects. Their perturbative expressions are given in Fig. 4. All $K^{(n)}$ are static interactions that cannot be turned off. We illustrate their influence on short timescales by considering the evolution of a coherent state $|a\rangle$ centered on $\bar a = \langle a|\hat a|a\rangle$. Using a semi-classical approximation, exact for coherent states, and that $|a|^2$ is a constant of motion, $\bar a$ rotates at the rate

$$\arg\bar a(t) = -\left(\omega_r + \sum_{n=1}^{\infty}K^{(n)}|a|^{2n}\right)t. \quad (11)$$

The $|a|^2$ dependence implies that a superposition of coherent states will deform under free evolution. This clearly undermines our control of the system. The main feature of the SNAIL dipole is the possibility to tune the $K^{(n)}$ coefficients via $\phi_e^{dc}$. We make use of this control knob to cancel the static nonlinearity, at least in the $|a|^2 \lesssim 1$ region, by imposing $K^{(1)}(\phi_e^{dc}) = 0$.

We experimentally locate this Kerr-free flux point by the out-and-back protocol[9], in which we monitor the drift angle $\theta$ of a coherent state of amplitude $a$ under a $t = 100$ ns free evolution. To do so, we select the $\theta$ value that maximizes the vacuum overlap of the state after free evolution and displacement by $-|a|\exp(-i\theta)$, as sketched in Fig. 5 (inset). The free evolution time, here $t = 100$ ns, is chosen short enough so that the state stays close to a coherent state. Each displacement pulse lasts 10 ns. The result, shown in Fig. 5, demonstrates a Kerr-free zone in phase space expanding up to $|a| \simeq 1.5$ for $\phi_e/\Phi_0 = 0.393$. By

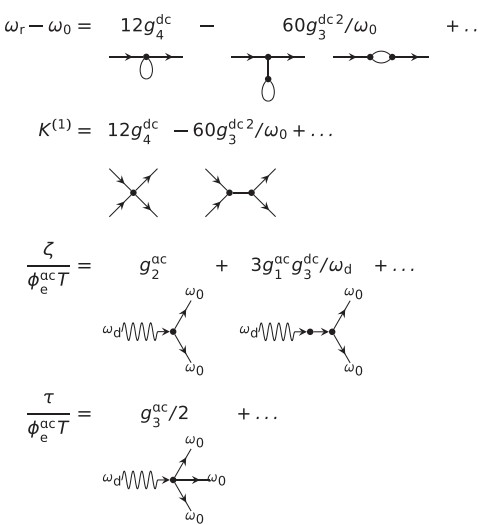

$$\omega_r - \omega_0 = \quad 12g_4^{dc} \quad - \quad 60g_3^{dc\,2}/\omega_0 \quad + \ldots$$

$$K^{(1)} = \quad 12g_4^{dc} \quad -60g_3^{dc\,2}/\omega_0 + \ldots$$

$$\frac{\zeta}{\phi_e^{ac}T} = \quad g_2^{ac} \quad + \quad 3g_1^{ac}g_3^{ac}/\omega_d \quad + \ldots$$

$$\frac{\tau}{\phi_e^{ac}T} = \quad g_3^{ac}/2 \quad + \ldots$$

**Fig. 4 | Perturbative expansions of static and driven nonlinear processes.** Each term is represented by a representative diagram, in spirit of[51,58,59], which aids in exhausting all contributions. A vertex with n straight legs represents $g_n$. An extra wavy line indicates ac (driven) terms. Incoming (resp. outgoing) straight line represents $\hat{a}$ (resp. $\hat{a}^\dagger$) in the represented process, which must conserve energy between drive, incoming and outgoing photons to be resonant. $T$ represents the effective gate duration.

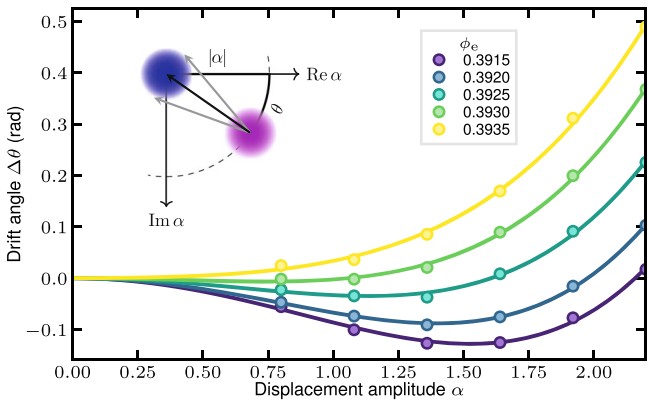

**Fig. 5 | Calibration of the Kerr-free operation point.** Drift angle measured as a function of $|\alpha|$, for several $\phi_e^{dc}$ values after 100 ns free evolution. Solid lines indicate fits according to Eq. (11), and stars indicate the measured drifted angles. (inset) Sketch of the drift angle measure protocol. The vacuum is displaced by $|\alpha|$ and freely evolves, accumulating a phase $\theta$ (pink). We then search for the displacement that maximizes overlap with the vacuum state (blue), whose angle is $\theta$.

fitting the drift angle using Eq. (11), we obtain the experimental value of $K^{(1)}$ plotted in Fig. 1f, which agrees with the perturbative expansion prediction, calculated from $g_n^{dc}$ which values were obtained independently from the fitting procedure described above.

## Calibration of effective drive strengths
When driving the system, the RWA approach is still valid, but non-number conserving terms appear in the effective Hamiltonian:

$$-iT\hat{H}_{eff,drive} = \left( \alpha\hat{a}^\dagger + \frac{\zeta}{2}\hat{a}^{\dagger2} + \tau\hat{a}^{\dagger3} + \ldots \right) - h.c., \quad (12)$$

which correspond to displacement, squeezing, and trisqueezing respectively. Their expression in terms of elementary processes is given in Fig. 4. The effective gate time $T = \int_0^t f(\tau)d\tau$ where $t$ is the total gate time and $f(\tau)$ is the pulse shape. Note that the diagrams represent

each process' resonance condition: $\omega_d = \omega_0$ for displacement, $2\omega_0$ for squeezing, and $3\omega_0$ for trisqueezing.

Since the different drive lines have different amplification, attenuation, and potentially unwanted reflectance, it is necessary to calibrate the drive amplitudes $\phi_e^{ac}$ independently at each frequency multiple. We measure $\alpha$, $\zeta$ or $\tau$ via Wigner tomography after flux driving at the corresponding frequency for a range of flux-pulse voltage amplitudes $V^{ac}$. The measurement settings are replicated in simulations and the linear scaling factor $V_0$ to match the corresponding flux amplitude $\phi_e^{ac} = V^{ac}/V_0$ is fitted for each measurement of $\alpha$, $\zeta$ or $\tau$, respectively, as shown in Fig. 2c, f. Note that the simulation and the experiment still match beyond the linear regime at low $\phi_e^{ac}$. This calibration is especially crucial to match the amplitude of the $1\omega_0$ and $3\omega_0$ in the cubic state protocol below.

## Experimental cubic phase state calibration
Experimental generation of the cubic phase gate requires a precise experimental tune-up to balance the timing, strength, and angles of the drive pulses. All pulses in this work are produced by direct digital synthesis in the Presto platform[53] up to 9 GHz. The $3\omega_0$ pulse at 12.5 GHz is produced by combining the output from two Presto ports in a physical mixer. Hence, we have full phase control of all pulses. The cubic gate, $\exp(i\gamma((\hat{a}^\dagger + \hat{a})/\sqrt{2})^3)$, contains terms resonant at $3\omega_0$ as well as cross terms like $\hat{a}^{\dagger2}\hat{a}$ requiring a drive at $1\omega_0$. To activate the interactions with equal strength, the effective drive amplitudes $\phi_e^{ac}$ are calibrated according to the procedure in the previous section. However, such a $1\omega_0$ flux drive also triggers a displacement via $g_1^{ac}$. It can be counterbalanced by charge driving, also at $1\omega_0$, with opposite phase and equal amplitude. This cancellation must be carefully realized, since $g_1^{ac}/g_3^{ac} \approx 2100$, a small misalignment would generate a significant displacement, well outside the Kerr-free zone. The amplitudes of the drives at $1\omega_0$ are first calibrated separately for charge and flux by displacing the vacuum with different amplitudes. We then fit the corresponding Poissonian distributions obtained by probing the qubit with a Fock-0 selective $\pi$-pulses to obtain individual amplitude scalings for the flux and charge drives. We then characterize the timing of the two $1\omega$ pulses down to ~100 ps, by utilizing the digital delay of the arbitrary waveform generator (see Supplementary Note 3 for details). Finally, we finetune the angle and magnitude of the flux drive such that under both drives, the vacuum stays undisplaced. Angles of the squeezing and trisqueezing drives are calibrated by applying each process separately and fitting the angle of the resulting Wigner states. With all amplitudes, angles, and timings calibrated, we apply the gate sequence of the main text to generate the cubic phase state.

## Error analysis
Measurements and simulations (as described above) are made to obtain the infidelity analysis presented in Fig. 6, where each bar corresponds to a case with a certain infidelity channel removed. The infidelity in each case is defined as the infidelity with respect to the ideal cubic state whose $\zeta$ and $\gamma$ parameters maximize fidelity to the simulated state. The error budget is found to be non-additive and nonlinear. For example, setting $g_n = 0$, $n \geq 5$ does reduce the infidelity if both thermal and dephasing are eliminated first. Note that $g_3^{dc}$ and $g_4^{dc}$ also generate contributions on higher orders. The error bar of the measured infidelity is obtained by dividing the data into 11 chronological batches (each >1 h of averaging), reconstructing each batch, and calculating the fidelity with respect to the best fit for all averaged batches. Part of the measured infidelity is found to originate from the drift of the system which is observed as a slightly tilting cubic phase state when the system slightly deviates from the Kerr-free point. Note that the slow drift of the dc-flux offset is not considered in the simulation but is an important factor in the difference between the collected data batches.

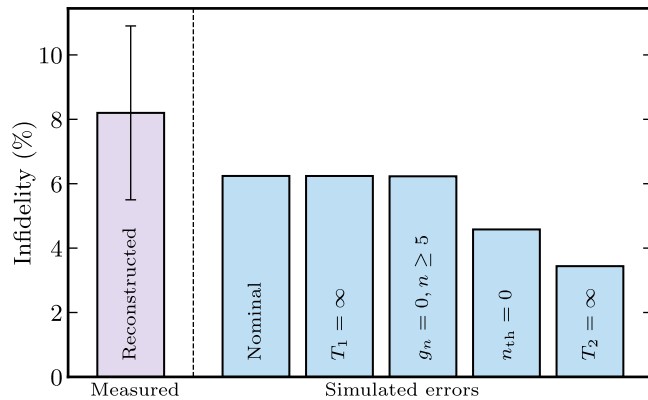

**Fig. 6 | Simulated error analysis showing how the infidelity is reduced if an error source is eliminated.** Simulated nominal infidelity is obtained by numerical simulation as described in the Section "Numerical simulations", with all parameters at their measured values.

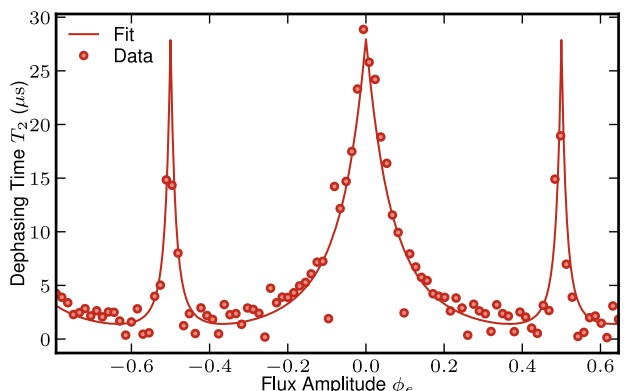

**Fig. 7 | The cavity $T_2$ measured by Ramsey interference protocol as a function of external flux.** The data are fitted by a model including $T_1$, broadband, and $1/f$ flux noise.

## Dephasing noise

The main source of decoherence for our system is flux noise, which induces fluctuations in the resonator frequency, and thus dephasing. According to this noise mechanism, the $T_2$ time should strongly depend on the $\omega_0$ dependence w.r.t. $\phi_e$. Following[54,55], we assume the flux noise power spectrum to be composed of a $1/f$ part and a broadband part, $S_{\phi_e}(\omega) = 2\pi A_{1/f}/\omega + S_{bb}$, which lead to

$$\frac{1}{T_2} = \frac{1}{2T_1} + \sqrt{2\log(2)A_{1/f}}\frac{d\omega_0}{d\phi_e} + S_{bb}\left(\frac{d\omega_0}{d\phi_e}\right)^2. \quad (13)$$

The fit to experimental data (see values in Supplementary Table 1) in Fig. 7 shows good agreement with this model. One notes the existence of sweet spots protected from dephasing at $\phi_e = 0$ and $1/2$. Since they do not coincide with the Kerr free flux point found at $\phi_e \simeq 0.3930$, they cannot be exploited in our current device. Optimizing the sweet spot position to enhance $T_2$ is a promising avenue to enhance the fidelity of any operation.

## Device fabrication

The device is fabricated from e-beam evaporated Aluminium on a 330 μm thick *C*-axis sapphire. The large features are patterned with an optical lithography and followed by a wet etch. The junctions are added with e-beam lithography and attached to the base layer with a separate patch layer[56]. The SNAIL has three junctions in series in one arm and one single junction in the other arm. The junctions are

fabricated with a variation of the Manhattan technique that allows for arbitrarily large junctions[57]. Based on a room temperature resistance measurement of similar junctions, the critical current of a large junction is 434 nA, while it is 46 nA for the small junction giving an asymmetry factor $\beta = 0.106$, close to the fitted value of 0.097.

## Data availability

The Wigner and spectroscopy data acquired in this study are available in the Figshare database under the accession code https://doi.org/10.6084/m9.figshare.25029041.

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

## Acknowledgements

A.M.E. would like to thank Giulia Ferrini for valuable discussions. The simulations and visualization of the quantum states were performed using QuTiP. The chips were fabricated in the Chalmers Myfab cleanroom. This work was supported by the Knut and Alice Wallenberg Foundation via the Wallenberg Centre for Quantum Technology (WACQT) and by the Swedish Research Council. T.H. also acknowledges the financial support from the Chalmers Excellence Initiative Nano.

## Author contributions

A.M.E. and M.Ke. built the experimental setup. A.M.E. carried out the experiments. M.Ke. and C.C.M. fabricated the chip. T.S. and T.H. developed the theory. A.M.E. and T.S. analyzed the data. A.M.E. and S.D. developed the measurement software framework. Y.L. and J.Y optimized and characterized an earlier version of the chip. T.S., M.Ke, T.H., and M.Kh. performed numerical simulations. A.M.E., T.S., T.H., M.Ke., M.Ku., M.Kh., P.D., and S.G. regularly discussed the project and provided insights. S.G. supervised the project. A.M.E., T.S., T.H., M.Ke., and S.G. wrote the paper with feedback from all authors.

## Funding

## Competing interests

The authors declare no competing interests.
