## [Peer Review File · Nature Communications]

Universal control of a bosonic mode via drive-activated native cubic interactionsREVIEWER COMMENTS

Reviewer #1 (Remarks to the Author):

The manuscript “Universal control of a bosonic mode via drive-activated native cubic interactions” by Eriksson et al. demonstrates universal control of a bosonic mode composed of a superconducting nonlinear asymmetric inductive element (SNAIL) resonator, enabled by native nonlinearities in the SNAIL element.

Topic of continuous-variable quantum computing is exceptionally relevant. The manuscript provides a novel experimental approach, advancing the field of continuous-variable quantum computing in microwaves. Technically, utilization of SNAIL – element allows to minimize Kerr nonlinearity effects while maintaining cubic phase state generation. The manuscript is very well organized. The results are very well represented, neat and are in a good agreement with numerical simulations/theory. I have a few comments to the authors, and I recommend paper for publication in Nature Communications.

Comment #1:

From abstract: “Our results initiate the experimental field of polynomial continuous-variables quantum computing.” The term “polynomial” seems to be new and followed from implementation of polynomial gate set (of orders from 0 to 3) in SNAIL-resonator. I think, if this term is describing a new path of CV quantum computing, there should be more discussion about it, why is it important. If it was in literature before, please provide the links.

Comment #2:

The paper focused on experimental generation of the cubic phase gate, to show completeness and universality of gate set for quantum computing. Work has been done without ancilla control qubit.

1. Maybe in text it worth to mention one more article of quantum computing with ancilla control qubit ([Ma, W. L., Puri, S., Schoelkopf, R. J., Devoret, M. H., Girvin, S. M., & Jiang, L. (2021). Quantum control of bosonic modes with superconducting circuits. *Science Bulletin*, 66(17), 1789-1805.]).
2. Also, authors claim, that in mentioned reference [33] (Heeres, R. W. et al. Implementing a

universal gate set on a logical qubit encoded in an oscillator. Nature communications, 8(1), 94.) control pulses are slower 10-100 times than in current manuscript. However, energy relaxation time in reference [33] is ~ 100 times longer: 2.7 ms compared to 28us in manuscript. Dephasing time in [33] is 43us compared to 2.8us (Kerr-free point) in manuscript. Despite on short duration of applied control pulses (to create a cubic phase state for example), T1 and T2 seem to be low. That must restrict number of gate operations. Any comments T1 T2 times can be improved?

Comment #3:

While working with Presto, which has high-frequency cutoff 9 GHz, to create signal on 3w_0, 2 output channels are in use: 1st creates 7.9 GHz LO for external mixer, second channel controls the pulse shape. Second channel has center frequency $3 \times 4.15 - 7.9 = 4.55$ GHz. IF port of mixer ZX05-153-S+ is limited by cutoff frequency of 4GHz (from datasheet). Perhaps, utilization of mixer in regime out of "borders" create unwanted mixing products in resulted frequency of 3w = 12.45 GHz. You are using quite broadband filters after mixer (9-13GHz at S1 figure in Supplementary Material) . How "correct" signal at 3w frequency? Maybe that is the factor as well, which contributes to decreasing of fidelity of cubic phase gate? Please, provide some comments.

Also, would be good to mark channels of Presto on S1 figure, which are used to generate single, double or triple frequency. Now its clear only from labels on filter symbols.

Small comment #4:

-Small typo (comma at row 31) "efficient manner [3, 6–10], or ,in principle, implement".

Reviewer #2 (Remarks to the Author):

The manuscript is devoted to the experimental implementation of a set of single-mode operations on a superconducting platform using the nonlinear SNAIL element. These included the non-Gaussian operations of trisqueezing and cubic phase gate. Both types of operations belong to cubic transformations and can be used to implement a universal set of single-mode operations. Previously, the authors in [M. Kudra et al., PRX Quantum 3, 030301 (2022)] demonstrated experimentally the generation of non-Gaussian states, including the

cubic phase state. However, the experimental implementation of a non-Gaussian gate is a much more difficult task. The authors were able to compensate for the always-on Kerr evolution, which is a difficult-to-separate process preventing the implementation of the required transformations in superconducting systems.

Using a cubic phase gate, the authors generated a cubic phase state. The result of this procedure can be compared with the result of the previous work of the authors [M. Kudra et al., PRX Quantum 3, 030301 (2022)]. The authors demonstrate deterministic generation of the cubic phase state, and the method proposed by the authors has a number of advantages over the previously proposed one: the generation time of the non-Gaussian state is shorter than before on the order; the degree of nonlinearity can be controlled by the interaction time. These are very significant achievements, so the authors' work can be called breakthrough.

The experimental implementation of non-Gaussian transformations presented in this work will be of interest to a wide range of readers working both in the field of quantum computing and in related fields.

I have a few minor comments that might improve the readability of the manuscript:

1) To facilitate readability, the structure of the presentation should be slightly changed. In particular, before giving expressions for the implemented set of gates (2)-(5), it is worth giving an explicit form of the effective Hamiltonian (12). This will make the pass from the full Hamiltonian of the system to the realizable transformations more obvious, especially for readers who do not work directly with these kinds of systems.

2) In the introduction it is worth revealing more widely the possibilities of using non-Gaussian operations. For example, a cubic phase gate can be used not only to obtain a universal set of single-mode operations, but to implement a T-gate on oscillators encoded using GKP states [D. Gottesman, A. Kitaev, and J. Preskill, Phys. Rev. A 64, 012310 (2001)], to improve the accuracy of state teleportation in continuous variables [E.R. Zinatullin, S.B. Korolev, and T.Yu. Golubeva, Phys. Rev. A 104, 032420 (2021)], as well as for the generation of other non-Gaussian states, such as the Schrödinger's cat state [I.V. Sokolov, Physics Letters A 384(29), 126762 (2020)]. I recommend that authors add these links to the introduction of the article.

Thus, the article can be published with minor revisions.

We thank the reviewers for their time, effort and comments which have helped us to improve our manuscript. Below follows a one-to-one list of responses to the review comments and the improvements we have made to the paper.

Reviewer #1

Comment #1:

From abstract: "Our results initiate the experimental field of polynomial continuous-variables quantum computing." The term "polynomial" seems to be new and followed from implementation of polynomial gate set (of orders from 0 to 3) in SNAIL-resonator. I think, if this term is describing a new path of CV quantum computing, there should be more discussion about it, why is it important. If it was in literature before, please provide the links.

Authors reply:

The term 'polynomial' refers to the notion introduced by Lloyd and Braunstein in their seminal paper. In contrast to e.g. achieving universal control via a qubit, which corresponds to gates non-trivially related to the creation and annihilation cavity operators. Here, we demonstrate gates which are polynomial in these operators, as originally proposed, which natively implement, for example, the cubic phase gate.

Changes to the manuscript:

End of abstract, we replaced

"Our results initiate the experimental field of polynomial continuous-variables quantum computing."
with

"Our results initiate the experimental field of polynomial quantum computing, in the continuous-variables notion originally introduced by Lloyd and Braunstein."

Comment #2:

The paper focused on experimental generation of the cubic phase gate, to show completeness and universality of gate set for quantum computing. Work has been done without ancilla control qubit.

1. Maybe in text it worth to mention one more article of quantum computing with ancilla control qubit ([Ma, W. L., Puri, S., Schoelkopf, R. J., Devoret, M. H., Girvin, S. M., & Jiang, L. (2021). Quantum control of bosonic modes with superconducting circuits. Science Bulletin, 66(17), 1789-1805.]).

Authors reply:

The article by Ma et al. is mentioned in the introduction

"However, to operate a CV mode, it is crucial to introduce a nonlinearity that allows one to universally manipulate the superpositions of the quantum states [4]"

and we refer to several examples where groups have used ancilla control qubits in

"These CV modes can be constituted by superconducting cavities weakly (dispersively) coupled to a qubit providing the universal controllability~\cite{hofheinz2009, heeres2015, Kudra2022,eickbusch2022}."

2. Also, authors claim, that in mentioned reference [33] (Heeres, R. W. et al. Implementing a universal gate set on a logical qubit encoded in an oscillator. Nature communications, 8(1), 94.) control pulses are slower 10-100 times than in current manuscript. However, energy relaxation time in reference [33] is ~100 times longer: 2.7 ms compared to 28us in manuscript. Dephasing time in [33] is 43us compared to 2.8us (Kerr-free point) in manuscript. Despite on short duration of applied control pulses (to create a cubic phase state for example), T1 and T2 seem to be low. That must restrict number of gate operations. Any comments T1 T2 times can be improved?

Authors reply:

We agree that the lifetimes of our planar circuit are considerably shorter than the ones typically achieved by 3D cavities. Yet, this first proof of principle has well-identified avenues for improvement. We extended the existing discussion on how to improve coherence time by the possibility to investigate a better trade off in ATS elements.

Changes to the manuscript:

Page 5 in Discussion, we added

"Another alternative is to optimize the tradeoffs in an ATS element^{~\cite{miano2021}}. An improved fabrication process is expected to extend the cavity lifetime^{~\cite{biznarova2023}}."

Comment #3:

While working with Presto, which has high-frequency cutoff 9 GHz, to create signal on 3w_0, 2 output channels are in use: 1st creates 7.9 GHz LO for external mixer, second channel controls the pulse shape. Second channel has center frequency $3 \times 4.15 - 7.9 = 4.55$ GHz. IF port of mixer ZX05-153-S+ is limited by cutoff frequency of 4GHz (from datasheet). Perhaps, utilization of mixer in regime out of "borders" create unwanted mixing products in resulted frequency of $3w = 12.45$ GHz. You are using quite broadband filters after mixer (9-13GHz at S1 figure in Supplementary Material) . How "correct" signal at 3w frequency? Maybe that is the factor as well, which contributes to decreasing of fidelity of cubic phase gate? Please, provide some comments.

Also, would be good to mark channels of Presto on S1 figure, which are used to generate single, double or triple frequency. Now its clear only from labels on filter symbols.

Authors reply:

We agree with these reflections. When analysing the mixer output with a spectrum analyser, we indeed see unwanted frequency content (combinations of Presto output and the fact that the mixer generates both sidebands), which we try to suppress with filters.

Changes to the supplementary materials:

In the first paragraph, we added:

" We note that the mixer is used outside of its specified IF bandwidth, which goes up to 4000 MHz,, while we use an IF of around 4574.6 MHz. Investigating the resulting frequency content in a spectrum analyser showed appropriate mixing. However, the Nyqvist-band outputs from the Presto being mixed in a three-port mixer results in undesired frequency peaks, which we try to suppress with filters. More narrow band filtering could improve the ω_0 pulse generation and potentially improve the fidelity of the applied gates."

We updated Fig S1 to mark which Presto ports are used for which pulse as well as the symbol for the DC block.

Small comment #4:

-Small typo (comma at row 31) "efficient manner [3, 6–10], or ,in principle, implement".

Authors reply:

Thanks, corrected!

Changes to the supplementary materials:

replaced "or ,in" with "or, in"

Reviewer #2 (Remarks to the Author):

1) To facilitate readability, the structure of the presentation should be slightly changed. In particular, before giving expressions for the implemented set of gates (2)-(5), it is worth giving an explicit form of the effective Hamiltonian (12). This will make the pass from the full Hamiltonian of the system to the realizable transformations more obvious, especially for readers who do not work directly with these kinds of systems.

Authors reply:

We acknowledge that it may not be obvious how the different interactions to implement the gates in (2-5) are engaged given our physical Hamiltonian. To clarify, we have tried to bridge the explanation of how the different interactions are engaged by connecting better to the Hamiltonian.

Changes to the manuscript:

Page 3 few sentences below Eq. 5, we added

"The ability to engage these processes from the Hamiltonian in equation (\ref{eq:hamiltonian}) can be intuitively understood by the drive field $f(t) \propto \cos(n\omega_0 t)$, resonantly capturing interactions \hat{a}^n and $\hat{a}^{\dagger n}$ at different orders n ."

2) In the introduction it is worth revealing more widely the possibilities of using non-Gaussian operations. For example, a cubic phase gate can be used not only to obtain a universal set of single-mode operations, but to implement a T-gate on oscillators encoded using GKP states [D. Gottesman, A. Kitaev, and J. Preskill, Phys. Rev. A 64, 012310 (2001)], to improve the accuracy of state teleportation in continuous variables [E.R. Zinatullin, S.B. Korolev, and T.Yu. Golubeva, Phys. Rev. A 104, 032420 (2021)], as well as for the generation of other non-Gaussian states, such as the Schrödinger's cat state [I.V. Sokolov, Physics Letters A 384(29), 126762 (2020)]. I recommend that authors add these links to the introduction of the article.

Authors reply:

We are encouraged by the reviewer's suggestions to expand on more applications of the implemented gates. We already refer to Gottesman et al. (2021), to not exceed the recommended numbers of references too much, we have added the following:

Changes to the manuscript:

Page 1 last sentence of paragraph 2 in Introduction, we added a reference to [E.R. Zinatullin, S.B. Korolev, and T.Yu. Golubeva, Phys. Rev. A 104, 032420 (2021)]

Additional changes to the manuscript:

For clarity, we also changed the symbol for the readout frequency to ω_{ro} in the last row of Table 1 in the supplementary.

We added headings and subheadings according to Nat. Comm.'s guidelines and updated the referencing to the Supplementary notes accordingly.

Added 'Data availability' paragraph

Corrected typo "controlability" to "controllability"

Fig3d added "with 1 std. dev. error bars"

We added the dc and ac coefficients to Table 1 in Supplementary Information.